# Nitrogen Source Affects the Composition of Metabolites in Pepper (*Capsicum annuum* L.) and Regulates the Synthesis of Capsaicinoids through the GOGAT–GS Pathway

**DOI:** 10.3390/foods9020150

**Published:** 2020-02-05

**Authors:** Jing Zhang, Jian Lv, Jianming Xie, Yantai Gan, Jeffrey A. Coulter, Jihua Yu, Jing Li, Junwen Wang, Xiaodan Zhang

**Affiliations:** 1College of Horticulture, Gansu Agricultural University, Yingmeng Village, Anning District, Lanzhou 730070, China; Jingzhanggs2019@outlook.com (J.Z.); lvjian@gsau.edu.cn (J.L.); yujihua@gsau.edu.cn (J.Y.); lj@gsau.edu.cn (J.L.); wangjwgau@outlook.com (J.W.); zhangxiaodan1993@outlook.com (X.Z.); 2Agriculture and Agri-Food Canada, Swift Current Research and Development Centre, Swift Current, SK S9H 3X2, Canada; yantai.gan@canada.ca; 3Department of Agronomy and Plant Genetics, University of Minnesota, St. Paul, MN 55108, USA; jeffcoulter@umn.edu

**Keywords:** ammonium, placenta, liquid chromatography–mass spectrometry, capsaicinoid synthetase, glutamine synthetase, phenylalanine ammonia-lyase

## Abstract

Phytochemical analyses of pepper fruit metabolites have been reported; however, much less is known about the influence of different forms of nitrogen (N), which is critical for plant growth and fruit quality formation. The “Longjiao No. 5” variety (*Capsicum annuum* L.) grown in Northwestern China was profiled using liquid chromatography–mass spectrometry (LC–MS) coupled with multivariate data analysis to explore the composition of different metabolites in pericarp and placenta, and to investigate the effect of three ammonium (NH_4_^+^) to-nitrate (NO_3_^−^) ratios (0:100, 25:75, and 50:50). A total of 215 metabolites were obtained by qualitative analysis, where 31 metabolites were the major differential metabolite components of pepper fruits between placenta and pericarp, and 25 among N treatments. The addition of ammonium up-regulated carbohydrates, such as Î±-lactose and sucrose, as well as phenylalanine lyase (PAL) of placenta tissue. The supply of 25% NH_4_^+^–N and 75% NO_3_^−^–N exhibited a relatively higher levels of ascorbic acid in pericarp and amino acids, capsaicin, and dihydrocapsaicin in placenta, and led to higher fruit weight among the ammonium-to-nitrate ratios. The expression and activities of glutamic acid synthetase (GOGAT) and glutamine synthetase (GS) that are involved in ammonium assimilation were affected by adjusting the ammonium–N proportion, and they were significantly positively correlated with capsaicin, dihydrocapsaicin contents, capsaicinoid synthetase (CS), as well as the relative expression levels of genes related to capsaicinoid biosynthesis, such as acyltransferase 3 (*AT3*) and acyl-ACP thioesterase (*FatA*).

## 1. Introduction

Nitrate (NO_3_^−^) and ammonium (NH_4_^+^), as the two main soil nitrogen (N) sources available to plants, exert different effects on biochemical processes in higher plants. When plants are supplied with nitrate, they first reduce the nitrate to ammonium through two energetically expensive steps of N assimilation, namely those associated with nitrate and nitrite reductase [1]. Assimilation of ammonium from root absorption, biological fixation, nitrate reduction, and photorespiration is largely accomplished by the glutamine synthetase (GS)–glutamate synthetase (GOGAT) pathway. In this pathway, GS combines glutamic acid with NH_4_^+^ to form glutamine, and the increase of glutamine level in plants stimulates the activity of GOGAT [2]. For the past few years, a large number of studies have focused on the utilization of NO_3_^−^ and NH_4_^+^ fertilization [3,4] and the effects on plant growth and development, such as plant photosynthetic physiology [5], rhizosphere environment [6], and abiotic stress tolerance [7]. However, few studies have been conducted to evaluate the relationship between N sources and agricultural product quality. With global improvement in human living standards, consumers demand for high-quality and nutritious agricultural products has increased.

Many studies revealed that the metabolism from the pericarp and placental tissues of Solanaceae fruits is different. A study on tomato (*Lycopersicon esculentum* cv. Micro–Tom) carbohydrate metabolism indicated that soluble sugars accumulate less in the placenta in comparison with the pericarp, while starch is degraded faster in the pericarp [8]. Chen et al. determined the phenolic compounds contents and antioxidant activity in extracts from placenta, pericarp, and stalk in red pepper (*Capsicum annuum* L.), and found they were different among the three tested parts [9]. Liu et al. performed RNA-seq analyses of the placenta and pericarp from pepper (*Capsicum frutescens* L.) and found that more than 4000 genes had significantly different expression levels between the placenta and pericarp [10]. Therefore, we speculate that there are more different metabolites and nutrients in the placenta and pericarp of capsicum fruits, which will provide the basis for improving the nutritional value of capsicum fruits.

Pepper (*Capsicum* spp.), which originated in Mexico [11], is generally recognized as a rich source of numerous phytonutrients, such as vitamin C, carotenoids, carbohydrates, and secondary metabolites [12,13]. Additionally, pepper is a good source of natural perfume due to its unique spicy taste [14]. Capsaicinoids are the source of pungent flavor in fruits of capsicum, and its biosynthesis is mainly through the phenylpropanoid pathway from phenylalanine to vanillin and the branched fatty acid synthesis pathway from valine to 8-methyl-6-sunolate CoA [15]. Studies using tracer technology and electron density scanning microscopy showed that the capsaicinoids synthesis site was mainly formed in the vacuoles of the epidermal cells of fruit placenta, and then accumulated in the vacuoles of the epidermal cells of pericarp [16]. More than 20 different capsaicinoids have been reported in the literature, with both a wide range of applications in the food industry and various health benefits for humans [17,18], such as the pain-relieving effects [19,20], body fat reduction benefits [21], and anticancer prospects [22].

It has been reported in several studies that the synthesis of capsaicinoids is determined by genotype and greatly influenced by exogenous substances or the growing environment. Ravishankar found that the addition of putrescine (0.1 mmol L^−1^) promoted capsaicin production in capsicum cells by suspension culture [23]. Akladious and Mohamed reported that the combined application of humic acid and low concentration of calcium nitrate could increase the capsaicin content under salt stress [24]. Capsaicinoid contents of six chili cultivars were evaluated in six environments significant differences were found among cultivars and growing environments [25]. Some studies on *Capsicum chinense* Jacq. and *Capsicum baccatum* L have indicated that the synthesis of capsaicinoids is greatly influenced by soil conditions, particularly organic carbon content, microbial activity, and fertilizer content [26,27]. We hypothesize that the supply of N in different forms will also affect capsaicinoid synthesis in pepper.

Although China produced 17.8 million metric tons of chili pepper fruit in 2017 (United Nations Food and Agriculture Organization statistics, 2017) most studies on pepper in China have focused on yield [28] and resistance to diseases [29,30]. In comparison, relatively few studies have assessed the nutritional and functional properties of peppers and no study has explored the effect of ammonium- and nitrate–N ratios on capsaicinoids of pepper, and knowledge of the metabolome in pepper is still lacking. The related quality contributing traits and functional attributes of pepper are especially important for producers who continually strive to increase their profitability through management to meet the needs of consumers for high-quality vegetables. In this study, we compared three ammonium-to-nitrate ratios, the pericarp and placenta of fruits, and explored the regulatory factors of capsaicinoid biosynthesis in pepper at the metabolome level.

## 2. Materials and Methods

### 2.1. Plant Material and Growth Conditions

Pepper (*Capsicum annuum* L.) cultivar ‘Longjiao No. 5’ is a hot pepper hybrid developed by the Gansu Academy of Agricultural Science in Lanzhou, China, which is widely cultivated in Northwestern China. The experiment was carried out in a greenhouse at Gansu Agricultural University, Lanzhou, Gansu, China (N 36°05′39.86″, E 103°42′31.09″). The sterilized and germinating seeds were seeded in cave dishes. At the six true-leaf stage, seedlings with similar physical size were transplanted into plastic pots (25 cm in depth and 35 cm in mouth diameter) that were filled with 6 L of mixed media, consisting of quartz sand and vermiculite in a ratio of 3:1 (*v*:*v*), respectively. The conditions in the greenhouse were as follows: temperature of 28 ± 2 °C/18 ± 2 °C (day/night), photoperiod of 12 h, average photosynthetic active radiation at noon of 1000 μmol m^−2^ s^−1^, and relative humidity of 60%–70%.

For the treatments, three NH_4_^+^: NO_3_^−^ ratios (0:100, 25:75, and 50:50) were arranged in a complete randomized design with three replicates. The different treatments were started one week after transplantation. At the seedling, flowering, and fruiting stages, pepper plants were watered, respectively, with 500, 1000, and 1500 mL of three nutrition solutions per container at 6-d intervals. The ammonium–N in the nutrient solution was supplied by ammonium sulfate [(NH_4_)_2_SO_4_] and nitrate–N was supplied by potassium nitrate (KNO_3_) and calcium nitrate tetrahydrate [Ca (NO_3_)_2_·4H_2_O]. All plants received the same amount of N (10 mM), P (1 mM), K (6 mM), Ca (2.5 M), Mg (1 mM), and microelements throughout the experiment [31], as well as 7 μmol L^−1^ of nitrification inhibitor was added to each container to prevent nitrification of ammonium [32]. The pH of all the treatments was kept uniform and constant (6.6–6.8).

### 2.2. Steady-State Analysis of Polar Metabolites

Metabolite extraction: At commodity maturity (40 d after flowering), the fruits in three treatments were collected and divided into placenta and pericarp separately, and then stored immediately in a −80 °C refrigerator until further use. After homogenization in liquid N, 50 mg of freeze-dried pepper material was extracted by shaking with 800 µL of methanol supplied with 10 µL of internal standard dichlorophenylalanine (2.8 mg mL^−1^). The fruits were then placed in a tissue grinding machine (JXFSTPRP-24, Shanghai, China) and ground for 90 s at 65 HZ. After 30 min of water-bath and ultrasound in an ultrasonic system (PS-60AL, Shenzhen, China), the ground tissue was placed in −20 °C for 1 h. A polar metabolite fraction enriched for primary metabolites and small secondary compounds was obtained after centrifugation at 12,000 rpm for 15 min at 4 °C.

Liquid chromatography-mass spectrometry (LC-MS) detection: Metabolite profiling was conducted using a Ultimate 3000LC coupled with Q Exactive system (Thermo Fisher Scientific, Waltham, Germany), and three biological repetitions were analyzed for each group. Chromatographic separation was performed on a Hyper gold C18 column (2.1 mm × 100 mm × 1.9 μm) using mobile phase A (0.1% formic acid and 5% acetonitrile in deionized water) and mobile phase B (0.1% formic acid in acetonitrile). Mobile phase B was increased linearly from 0% at 0 min to 20% at 1.5 min to 100% at 9.5 min, and then held at 100% until 14.5 min. Finally, solvent B was decreased to 0% at 14.6 min and held at 0% until 18 min. The flow rate was maintained at 0.35 mL min^−1^. For ionization, positive and negative electrospray ionization modes were used. Scanning mode: Full Scan (*m*/*z* 70~1050) and data-dependent secondary mass spectrometry (dd-MS2, TopN = 10); Resolution: 70,000 (primary mass spectrometry) and 17,500 (secondary mass spectrometry). Collision mode: High energy collision dissociation (HCD) [33].

### 2.3. Fruit Weight and Capsaicinoids Determination

At 40 d after flowering, 12 fruits were randomly harvested from each treatment and washed with distilled water. The fresh weight was measured with an electronic balance, then both samples were divided into placenta and pericarp and dried separately at 55 °C in an electronic oven until reaching a constant weight. After weighing the dry pepper fruit, a grinder was used to crush samples into powder. Capsaicinoids were quantified with high-performance liquid chromatography (HPLC). For capsaicinoids extraction, 1 g (accurate to 0.0001) of pepper powder was extracted with 20 mL methanol-H_2_O (1:1, *v*:*v*) using a temperature-controlled ultrasonic machine SB25-12D (Ningbo Scientz Biotech Company, Ningbo, China) at 60 °C for three times. The extract solution was filtered through a millipore membrane of 0.22 μm pore size [34]. Capsaicin (Sigma-Aldich, St. Louis, MO, USA, 98%) and dihydrocapsaicin (Sigma-Aldich, St. Louis, MO, USA, 98%) were accurately weighted and dissolved in methanol with a concentration of 1 mg mL^−1^ and a series of solutions of 0, 20, 40, 60, 80, and 100 μg mL^−1^ were prepared with the mixed standard solution. 

The analysis system consisted of Waters Alliance HPLC system (Waters Corporation, Milford, America) equipped with 2998 PDA detector and a Symmetry C18 reverse-phase column (250 mm × 4.6 mm, 5 µm). The absorbance at 280 nm was measured using a Shimadzu UV-VIS 1700 spectrophotometer. A methanol:1% acetic acid aqueous solution (70:30) was used as the mobile phase with a flow rate of 1.0 mL min^−1^. The column temperature was 30 °C and injection volume was 5 µL.

### 2.4. Enzyme Activity

At 40 d after flowering, the fruits were collected, frozen separately in liquid N, and stored at –80 °C. The placenta and pericarp tissues were added to an appropriate amount of pre-cooled extracting solution for grinding into fine paste with a mortar and pestle by ice bath, then transferred into a centrifuge tube and centrifuged at 4 °C and 12,000× *g* for 10 min. The supernatant was used for the determination of the activities of phenylalanine ammonia-lyase (PAL) and capsaicinoid synthetase (CS) using a Plant PAL ELISA kit as well as Plant CS ELISA kit (Solarbio Science and Technology Ltd., Beijing, China) according to the manufacturer’s protocol. The reaction was terminated by the addition of a sulfuric acid solution, and the color change was measured spectrophotometrically at a wavelength of 450 nm. The concentrations of PAL and CS in the samples were then determined by comparing the absorbance value of the samples to the standard curve. 

For the determination of GS and GOGAT, tissues were ground to powder in a mortar that had been pre-cooled with liquid N and then homogenized in the extraction buffer (0.1 M phosphate buffer, pH = 7.5). The activity of GS was assessed using a GS kit (GS-2-Y) at 540 nm, and the GOGAT activity was determined by a GOGAT detection kit (GOGAT-2-Y) at 340 nm [35]. Absorbance was measured with a TU-1900 spectrophotometer (Beijing Persee Instruments Company, Beijing, China), and the sample content was calculated. The kits for analyzing enzyme activities were purchased from Comin Biotechnology Co. Ltd., Suzhou, China. Each measurement was performed with three biological replicates.

### 2.5. Quantitative RT-PCR

Total RNA of placenta and pericarp from three treatments were isolated from liquid N ground tissue using an RNA plant reagent (Tiangen, Beijing, China) according to the manufacturer’s protocol. Quality and concentration were checked on an Agilent 2100 spectrophotometer (Agilent Technologies, Foster City, CA, USA) and samples were stored at −80 °C until further use. gDNA Eraser (Tiangen, Beijing, China) was used to remove genomic DNA contamination, then a cDNA synthesis Kit (Tiangen, Beijing, China) was applied to synthesize cDNA. Finally, 1 μL of cDNA was diluted five-fold with sterile ddH_2_O and used for quantitative real-time PCR (qRT-PCR) analysis in a real-time PCR detection system (Lightcycler96 Real-Time PCR System, Roche, Basel, Switzerland). The reaction system includes: 1 μL of cDNA template, 10 μL of 2 × SYBR Green Master Mix (Tiangen, Beijing, China), and 0.8 μL of each primer, to a final volume of 20 μL by adding ddH_2_O. The two-step amplification procedure was used and consisted of one cycle of 95 °C for 15 min, denaturing 40 cycles of 95 °C for 10 s, then annealing at 60 °C for 20 s. Actin was used as reference genes; primers were used in this study for Phe ammonia-lyase (*PAL*), cinnamate 4-hydroxylase (*CH4*), caffeoyl-CoA 3-O-methyltransferase (*COMT*), Acyl-ACP thioesterase (*FatA*), acyltransferase 3 (*AT3*), and glutamine synthetase (*GS*) were described by Keyhaninejad et al. and Deng et al. [36,37]; and NADH-dependent glusynthase (*NADH-GOGAT*), ferrodoxin-dependent Glu synthase (*Fdx-GOGAT*), and putative aminotransferase (*AMT*) were designed by TakaRa Biotechnology Co., Ltd. (Dalian, China). The sequences used to design the primers are listed in Table 1. The values presented for each target gene were means of six biological replicates. We analyzed the relative gene expression using the comparative 2^−ΔΔCT^ method [38].

### 2.6. Statistical Analyses

Extraction and alignment of the LC–MS raw data were carried out using compound discoverer software (Thermo Fisher Scientific, Waltham, Germany) [39], which reorganized the data into a two-dimensional data matrix form, including the retention time, molecular weight, observation quantity (sample name), peak intensity, and other information [40]. To analyze differences between treatment groups, principal component analysis (PCA) and orthogonal projections to latent structures-discriminant analysis (OPLS-DA) on normalized data were performed in SIMCA (version 14.1, MKS Data Analytics Solutions, Umea, Sweden) [41,42]. OPLS-DA models were confirmed by a seven-fold cross validation, and a permutation test was further applied to rigorously validate the models’ reliability (*n* = 200). The combination of *p* value in Student’s *t*-test and variable importance in the projection (VIP) values from OPLS-DA were used as a coefficient for metabolite selection (VIP > 1.0 and *p* < 0.05). The data generated in fruit weight, capsaicinoids, enzymes, RT-PCR were subjected to Analysis of Variance (ANOVA) with the Tukey test (*p* < 0.05) to find statistically significant differences between the mean values; Pearson correlation coefficients were calculated for capsaicinoids, enzymes, and RT-PCR data integration using SPSS (version 15.0, SPSS Inc., Chicago, IL, USA).

## 3. Results

### 3.1. Overview of Metabolomic Profiling

The PCA score scatter plot of all samples including quality control (QC) samples is shown in Figure 1. The PC [1] and PC [2] represent the scores of the principal components ranked first and second respectively, all samples were within the 95% confidence interval (Hotelling’s t-squared ellipse). The PCA score plots revealed a clear separation among the different tissues and nitrogen treatments also; the biological replicates for each group (T1-PE, T2-PE, T3-PE, T1-PL, T2-PL, T3-PL) were always clustered together. It demonstrated high reproducibility of different ammonium-to-nitrate ratio treatments and tissues, as well as the significance of their effects on the metabolite levels of “Longjiao No. 5”.

A total of 215 compounds were qualitatively matched from the raw data of the analysis. The OPLS-DA analysis revealed that 39 differential metabolites were filtered (VIP > 1.0 and *p* < 0.05), which were responsible for 17.5% of carbohydrates and carbohydrate conjugates, 15% of amino acids, peptides, and analogues, and 7.3% of phenols and derivatives (Figure 2A). Parameters of the OPLS-DA model are shown in Appendix A. The prediction accuracy of models were good, since the *Q*^2^ of each model were all higher than 0.77. PLS-DA models were further confirmed by a seven-fold cross validation, and a permutation test was applied to validate the models’ reliability rigorously, the original model has no over fitting phenomenon, and the model has good robustness.

### 3.2. Differential Metabolites between Pericarp and Placenta

There were 15, 23, and 16 differential metabolites in T1, T2, and T3 between the pericarp and placenta (Figure 2B). The cluster heat map (Figure 3) shows that five compounds rich in pericarp were clustered together, while 32 compounds rich in placenta were clustered together. Under different N treatments, except for a few metabolites such as d-(-)-quinic acid and thiolactomycin, the majority of the metabolites showed significant differences between the pericarp and placenta. L-(-)-malic acid, gentiopicrin, gluconic acid, 2-linoleoyl-sn-glycero-3-phosphoethanolamine, lariciresinol 4-o-glucoside, and 1-caffeoyl-beta-d-glucose were common differential metabolites between placenta and pericarp tissue among three treatments. When supplied with both ammonium and nitrate, T2 and T3 had common differential metabolites such as 1-o-vanilloyl-beta-d-glucose, astragalin, melilotoside, Î±-lactose, asparagine, ascorbic acid, 4-oxoproline, fructosylglycine, l-histidine, d-(-)-glutamine. With T2 and T3, ascorbic acid was 0.03–0.04-fold greater in placenta than pericarp (Appendix A). Further differences of capsaicin and dihydrocapsaicin between the pericarp and placenta were also observed, and these were more pronounced under 25% NH_4_^+^-N and 75% NO_3_^−^-N.

### 3.3. Differential Metabolites among Treatments

The Venn diagram (Figure 2C) shows the number of metabolites accumulated in significantly different amounts among the three comparison groups. In total, 18, 10, and 9 differential metabolites were detected among T1 vs. T2, T1 vs. T3, and T2 vs. T3. From the results, the highest number of differentially occurring metabolites was found in T1 vs. T2. Asparagine was the only common differential metabolite among the three groups. Under both sole nitrate (T1) and two combinations of ammonium and nitrate (T2 and T3), the levels of most metabolites were changed in placenta tissue while only a few changes were observed in pericarp tissue. In the pericarp tissue, theophylline was the only compound up-regulated in T2 but down-regulated in T3, compared to T1; additionally, 5-aminolevulinic acid and gluconic acid were down-regulated in T2, and asparagine, theophylline, 4-oxoproline, and diethylpyrocarbonate were down-regulated in T3, compared with T1.

In placenta, 15 metabolites were increased in T2 compared with the sole nitrate supplied fruit (T1); three metabolites such as Î±-lactose and sucrose in T3 were higher than T1. However, avenein was significantly decreased when the fruits were supplied with two forms of N (T2 and T3). T3 supplied a higher level of ammonium (50%), but the differential metabolites in T1 vs. T3 were less than those in T1 vs. T2. In comparison with the treatment of sole nitrate–N, the addition of 25% ammonium–N significantly increased the level of vanilloids, including nonivamide, capsaicin, dihydrocapsaicin, and its glycosyl compound 1-o-vanilloyl-beta-d-glucose in placenta tissue by 17.42-, 5.44-, 5.63-, and 2.71-fold, respectively (Appendix A). In addition, among the 215 substances in this experiment, intermediate substances in capsaicinoids biosynthesis, such as l-phenylalaninein, cinnamic acid, p-coumaric acid, and vanillin, and another capsaicinoid-homodihydrocapsaicin has been identified (Figure 4).

### 3.4. Pathway Analysis of Metabolites

We used the Kyoto Encyclopedia of Genes and Genomes (KEGG) to sort the pathways (Raw *p* < 0.1) of differential metabolites and the number of metabolites in the pathway [43] (Table 2). Between the pericarp and placenta of pepper fruits, significant pathways (*p* < 0.05) were the biosynthesis of alkaloids derived from terpenoid and polyketide, carbon metabolism, taste transduction, and the phosphotransferase system. Among the three N treatments, 5-aminolevulinate, d-gluconic acid, and theophylline hit the pathway of microbial metabolism in diverse environments (*p* = 0.01), sucrose and lactose hit the pathway of carbohydrate digestion and absorption, as well as the pathway of galactose metabolism, d-gluconic acid hit the pentose phosphate pathway (*p =* 0.04), and l-asparagine hit the pathway of mineral absorption (*p =* 0.04).

### 3.5. Fruit Weight and the Quantification of Capsaicinoids in Pepper

Fruit weight analysis revealed that compared with the total nitrate–N treatment (T1), the supply of 25% ammonium–N (T2) significantly increased the fresh weight of pepper by 27.3% compared T1 (Figure 5A). Similarly, the dry weight of fruit increased by 48.0% compared with T1. However, when the amount of ammonium–N increased to 50% (T3), the fresh and dry weights of fruits decreased slightly, but there was no significant difference compared with T1 and T2.

In the validation of quantitative analysis using HPLC, the content of capsaicin (Figure 5B) and dihydrocapsaicin (Figure 5C) in pericarp of pepper were low, about 0.048–0.086 g kg^−1^ and 0.036–0.056 g kg^−1^, respectively. The contents of capsaicin (1.738–3.527 g kg^−1^) and dihydrocapsaicin (0.755–1.873 g kg^−1^) in the placenta of the three treatments were much higher than that of pericarp. The content of capsaicin was significantly affected by the proportion of ammonium and nitrate. The placenta treated with 25% ammonium and 75% nitrate (T2), contents of capsaicin and dihydrocapsaicin were 102.9% and 148.1% higher, respectively, than that of T1. Although the capsaicin content in pericarp of T2 increased, there was no significant difference compared with T1.

### 3.6. Analysis of PAL, CS, GOGAT, and GS Enzymes

By enzyme-linked immunoassay, PAL enzyme concentration in different tissues and treatments of pepper was determined (Figure 6A). In the pericarp, the addition of 25% ammonium–N (T2) and 50% ammonium–N (T3) had no effect on PAL enzyme concentration compared with 100% nitrate treatment (T1), but T2 and T3 were increased by 42.8% and 88.1%, respectively, in the placenta compared to T1. The concentration of CS in the placenta was significantly higher than that in the pericarp, and the concentration with T2 was significantly higher than that with T1 and T3 (Figure 6B).

The enzyme activities of GOGAT (Figure 6C) and GS (Figure 6D) in the pericarp of the three treatments were significantly lower than those in the placenta. For GOGAT, compared with T1, the enzyme activity of T2 was significantly increased in placenta and pericarp, but for GS, only T2 in the placenta was significantly higher than T1. For both enzymes, there was no significant difference in GOGAT and GS enzyme activities between T3 and T1.

### 3.7. Analysis of Gene Expression and Correlations

We performed quantitative RT-PCR analysis for selected gene sets to confirm the expression levels of ammonia metabolism and the genes relate to synthesis of capsaicin. The *PAL*, *C4H*, *COMT*, *BCAT*, *FatA*, and *AT3* as representatives of capsaicin biosynthetic genes were identified. The relative expression of these genes in the pepper pericarp were far lower than that in the placenta tissue. For *PAL* (Figure 7A) in both the pericarp and placenta, the relative expression levels in T2 and T3 were higher than that in T1. However, the results of *CH4* (Figure 7C) and *BCAT* (Figure 7D) showed that T2 and T3 were significantly lower than T1 in the placenta and there was no significant difference between T2 and T3, while there was no significant difference between the three treatments in the pericarp. The results of *COMT* (Figure 7B) and *FatA* (Figure 7E) showed that the addition of ammonium–N has an obvious effect on the expression of the two genes, and the highest relative expression was found in T2. The fluorescence quantitative results of *AT3* (Figure 7F) showed that the expression of T1 was low in the pericarp, while no expression was detected in T2 and T3. In the placenta tissue, the relative expression of AT3 in each treatment was high; T2 and T3 were 190.9% and 85.3% higher than T1, respectively. We also analyzed three ammonium metabolism genes, including *NADH-GOGAT* (Figure 7G), *Fdx-GOGAT* (Figure 7H), and *GS* (Figure 7I) and the relative expressions were similar with our capsaicin data. Compared with the total nitrate treatment (T1), the relative expression of *GS* in the placenta tissues of T2 and T3 was significantly increased, and was 64.2% and 51.1% higher than T1, respectively. The expression of T2 and T3 in pericarp was increased slightly, but there was no significant difference compared with T1. The change of *NADH-GOGAT* and *Fdx-GOGAT* in different tissues and different treatments was consistent. In the placenta tissue, the relative expressions of *NADH-GOGAT* and *Fdx-GOGAT* were significantly increased by T2 compared with T1, and there was no difference between T3 and T1.

The Pearson correlation of two kinds of capsaicinoids content, phenylalanine lyase, capsaicinoid synthetase, glutamate synthase, glutamine synthase, six structural genes of capsaicinoid synthesis, and three genes of the GS-GOGAT were analyzed (Figure 8). The results showed that correlation coefficients of capsaicinoid contents with CS, GOGAT, GS enzyme, and relative expression of *FatA*, *NADH-GOGAT*, *Fdx*- *GOGAT*, *GS*, and *AT3* genes were greater than 0.9, and were significantly correlated (*p* < 0.01). The correlation coefficient between CS and GS enzyme was up to 0.996 (*p* < 0.01). The relative expression of *AT3* was significantly correlated with not only the gene *FatA* and the concentration of CS, but also the relative expression of *NADH-GOGAT*, *Fdx*- *GOGAT*, and *GS* genes of nitrogen metabolism (*p* < 0.01). The concentration of PAL was significantly correlated with the relative expression levels of *PAL* and *GS* genes at *p* < 0.01, and was significantly correlated with dihydrocapsaicin content, capsaicinoid synthetase, glutamine synthase, and *GS*, *FatA*, *NADH-GOGAT*, and AT3 genes at *p* < 0.05.

## 4. Discussion

### 4.1. Metabolism of Pepper Pericarp and Placenta

The metabolism of pericarp and placenta of capsicum fruits was determined in this study, after irrigation with nutrient solutions composed of different N forms and ratios. More metabolites in placenta tissues changed significantly compared with pericarp tissues; therefore, the former may be more sensitive to N sources. Phenylalanine ammonia-lyase is widely found in plants and a few microorganisms and plays a vital role in the biosynthesis of plant secondary metabolites, in addition to participating in the biosynthesis of protein, flavonoid, lignin, and many other compounds [44]. In this study, PAL activity and *PAL* gene expression in placenta were higher than those in pericarp, and the N treatments markedly affected them in the placenta. This may be one of the reasons that more differential metabolites were concentrated in the placenta of pepper. Some substances that are involved in taste transduction, such as sucrose and l-malic acid, had higher values in the placenta. Although pericarp tissue accounts for most of the fruit, placenta tissue can provide more rich taste experience for pepper fruit, which is important for attracting consumers [45]. There are also some substances with high content in pericarp tissue, such as gluconic acid, ascorbic acid, astragalin. ascorbic acid, also known as vitamin C, is water soluble [46]. According to previous studies, capsicum is one of the richest known plant sources of vitamin C [11]. Among five *Capsicum annuum* cultivars, vitamin C content ranged from 63.1–64.9 mg 100 g^−1^ on a wet basis, and the cultivars that had higher ascorbic acid content had higher capsaicinoid contents as well [47]. Guil-Guerrero et al. reported that the vitamin C content of 10 pepper varieties ranged from 102–380 mg 100 g^−1^ and was higher than previously reported values [48]. Therefore, the consumption of pepper, especially pericarp, can supplement vitamin C needed by the human body and has great importance for human health.

As mentioned earlier, PAL is a key enzyme in the metabolism of phenylpropanoids; additionally, it is the first enzyme of capsaicinoid biosynthesis in the phenylpropanoid pathway. Perucka and Materska found that the content of capsaicin in capsicum fruit was positively correlated with the activity of PAL [49]; that is, the content of capsaicin begins to increase when high levels of PAL appeared. As expected, our results support Canto-Flick et al. [50] and Liu et al. [10], who reported that placenta has the highest content of capsaicinoids in pepper fruit, followed by pericarp. This is consistent with previous research reporting that the synthetic site of capsaicin is in the placenta [15]. However, there are some exceptions. Another study on the pepper variety ‘Trinidad Moruga Scorpion Yellow’ found that multiple capsaicinoid biosynthetic pathway genes were expressed exclusively in the pericarp, but in the pericarps of other spicy varieties, related genes are barely detected [51]. CS was believed to be the key enzyme in capsaicin synthesis and is located in the vacuolar membrane of the placental epidermal cells of pepper [52]. It catalyzes the final step of the capsaicin synthesis process; that is, the synthesis of capsaicin from 8-methyl-6-decenyl CoA and vanillinamine [53]. Immunoassay confirmed that *AT3* is specifically located in the placenta of capsicum fruit and it has been proposed to encode capsaicinoid synthetase [52,54]. In the present study, enzyme concentration of the CS and *AT3* genes in the placenta of capsicum fruit was much higher than that in the pericarp, which is consistent with a large number of previous studies [55].

### 4.2. Ammonium: Nitrate Ratio Affects Metabolites of Pepper

Nitrogen is one of the most important elements in plants, and a large number of studies have shown that different N forms have different effects on the growth and development of plants [56,57]. Research on apple treated with five NO_3_^−^:NH_4_^+^ ratios (2.5:0.1, 6:1, 6:0.7, 6:0.5, and 6:0.3) showed that fruit shape, juice pH, and dry matter were not affected significantly by NH_4_^+^ concentration; however, composition was influenced [58]. Compared to using NO_3_^−^–N as the sole N source, using 25% of NH_4_^+^–N in solution has been shown to produce higher tomato (*Lycopersicon esculentum* Mill.) yield [59]. With reference to pepper (*Capsicum annuum* L. cv. ‘California Wonder’), Marti and Mills showed that ammonium nutrition decreased pepper plant dry weight and fruit yield in comparison to solo nitrate; the nutrient contents of vegetative tissue were affected by three NH_4_^+^/NO_3_^−^ ratios (1:0, 1:1, and 0:1) and the stage of pepper development [60]. The total and high-quality pepper fruit yields both increased with decreasing NH_4_^+^/NO_3_^−^ ratio, as reported by Bar-Tal et al. [61]. Amino acids are an important part of the quality and flavor of agricultural products [62,63]. The metabolism of amino acids increases the protein synthesis process of basic metabolism, which is conducive to the synthesis and accumulation of protein in the fruit of pepper [64]. The present results showed that, in pepper placenta, the addition of ammonia–N up-regulated amino acids, peptides, and analogues, in particular asparagine, d-(-)-glutamine, l-histidine, aminolevulinic acid, n-acetylvaline, and fructosylglycine. The supply of 25% ammonia–N caused a significant increase compared to the other two treatments. This may due to the change of N source that facilitates the assimilation of ammonium–N in pepper. Once glutamic acid or glutamine is formed during the assimilation of ammonium–N, other amino acids are formed through the ammonia reaction. However, in sweet pepper fertilized with 100% NO_3_^−^–N and three forms of N (NO_3_^−^–N + NH_4_^+^–N + NH_2_–N), the concentration of free amino acids, soluble sugars, and dry matter in pepper fruits was unaffected by N form [65]. Metabolic products of polyunsaturated fatty acids have been variously implicated in control of microbial pathogens. 13(S)-HpOTrE, a monohydroperoxy polyunsaturated fatty acid, which was up-regulated in T2 and mainly found in the placenta tissue of the pepper, may take part in a lipid-based signaling system initiated by insect and pathogen attack [66].

The complex metabolic reactions and their regulation in pepper plants are not carried out alone. Because of the change of N sources, complex pathways and networks are often formed, and their interaction and regulation eventually leads to systematic changes in fruit metabolism [67]. The carbon and nitrogen metabolisms are tightly coordinated in plant [68,69]. The up-regulation of carbohydrate can be observed after the supply of ammonium–N. This is largely due to the assimilation of ammonium requiring a large amount of carbon skeleton; on the other hand, the process of carbon assimilation depends on the protein and enzyme from ammonium assimilation. Due to the similarity of metabonomics among organisms, we were able to better understand the physiological effects of metabolic substances on the human body. Sucrose and lactose, which are involved in the carbohydrate digestion and absorption pathway, were up-regulated in the placenta with the addition of ammonium–N to pepper. The galactose metabolism, phosphotransferase system, and pentose phosphate pathway that are related to sugar metabolism, were also significantly enriched with addition of ammonium–N. Asparagine, which is involved in the protein digestion and absorption pathway, as well as the mineral absorption pathways, was up-regulated when 25% or 50% of the N was supplied as ammonium. Plant secondary metabolites, which can be divided into three chemical groups (terpenes, phenolic compounds, and alkaloid) based on their biosynthetic origin [70], play a role in flavor substances originating from agricultural products. 5-Aminolevulinic acid, d-gluconic acid, and theophylline, affected by N treatment in the pericarp, were enriched in this secondary metabolites pathway with addition of ammonium–N, and also play a crucial role in plant defense against herbivores and pathogens [71]. However, there are substances that decrease with the increase of ammonium–N, such as d-(-)-quinic acid, thiolactomycin, and diethylpyrocarbonate.

### 4.3. Ammonia Nitrogen Addition Regulates Capsaicinoids Biosynthesis through the GOGAT/GS Cycle

Acyl-ACP thioesterase (FatA) catalyzes the termination of the fatty acid synthesis cycle and the removal of acyl-ACP to produce free fatty acids. Aluru et al. studied the relationship between composition factors of fatty acid synthase and pungency, and they found that transcription levels of *Kas*, *Acl*, and *Fat* were significantly positively correlated with pungency [72]. Results of fluorescence quantitative analysis showed that the expression trends of *FatA*, *AT3*, *NADH*–*GOGAT*, *Fdx*–*GOGAT*, and *GS* in the glutamate cycle were similar, and they were significantly correlated. NADH–GOGAT was located in the plastid of non–photosynthetic tissue, while Fdx–GOGAT was in the chloroplast, which is used for N metabolism of photorespiration. Ammonium assimilation is catalyzed by the GOGAT–GS pathway, possibly promoted by the addition of ammonium–N. with the T2 and T3 treatments. The correlation analyses indicate that the activity of CS was similar to GOGAT–GS in different treatments, and the spicy characteristics of pepper may be affected by the GOGAT–GS pathway. Previous work indicates that amino acids are primary metabolites and, therefore, are not included in the capsaicin synthesis pathway [73]. However, as reported by Mazourek et al., they fully described the pathways of the capsaicin biosynthesis model, which combines phenylpropanoid and benzenoid metabolism and medium–length, branched–chain fatty acid biosynthesis, and further includes branched-chain amino acid metabolism [74]. In the latter, glutamate and glutamine are thought to be precursors for fatty acid biosynthesis and the acyl moieties of capsaicinoids are derived from catabolism of amino acids with subsequent fatty acid elongation [75], thus leading to capsaicinoid biosynthesis.

Marti and Mills suggested that NO_3_^−^ should be at least 50% of the N form supplied to sweet pepper through fertilizer applications, although the amount of NH_4_^+^ supplied had no significant effect on N and P content of the fruit [60]. Our previous studies have shown that 25% of N applied as ammonium promoted the activity of GOGAT and GS, and the growth of pepper plants [76]; similar results have also been found in tomato [7]. However, compared with the total nitrate–N treatment, when the amount of ammonium–N was 50%, no significant changes of capsaicinoid content were observed. The addition of a high amount of ammonium–N. was not suitable for the optimal growth of pepper plants, and excessive levels of ammonium–N can be toxic to plants [77,78]. Therefore, a significant association of the increase of capsaicinoid content and the activities of GOGAT and GS enzymes in this experiment may be because its precursors, such as glutamate and glutamine were promoted by the addition of appropriate ammonium–N. Consequently, the present study found that the GOGAT and GS enzymes are important in the regulation of capsaicinoid biosynthesis and provides insights into the regulation of each capsaicinoid biosynthesis pathway. Regulation of the spiciness of pepper and quality of agricultural products by artificially optimizing N fertilization strategy remain future objectives.

## 5. Conclusions

Altogether, our metabolome data showed that the compounds in “Longjiao No. 5” pepper fruit were affected by the ratio of ammonium and nitrate, the metabolites in the placenta were more sensitive to the change of N source. As compared with the treatments of total nitrate–N, the addition of 25%–50% ammonium–N are beneficial to lactose, sucrose synthesis and phenylalanine lyase activity in placenta; the addition of 25% ammonium–N up-regulated the ascorbic acid in pericarp, similarly to amino acids, capsaicin, and dihydrocapsaicin in the placenta of pepper, however, they do not take advantage of the supply of 50% ammonium–N. The results also demonstrated a strong link between spiciness in pepper and proportion of N form, such as the content of two capsaicinoids and capsaicinoid synthetase. Additionally, genes related to capsaicinoid biosynthesis such as AT3 and FatA were positively correlated with glutamate synthetase and glutamine synthetase involved in ammonium assimilation. Adjusting both the N source and the proportion is an effective management approach to improving the nutrition and spicy characteristics of capsicum fruits.

## Figures and Tables

**Figure 1 foods-09-00150-f001:**
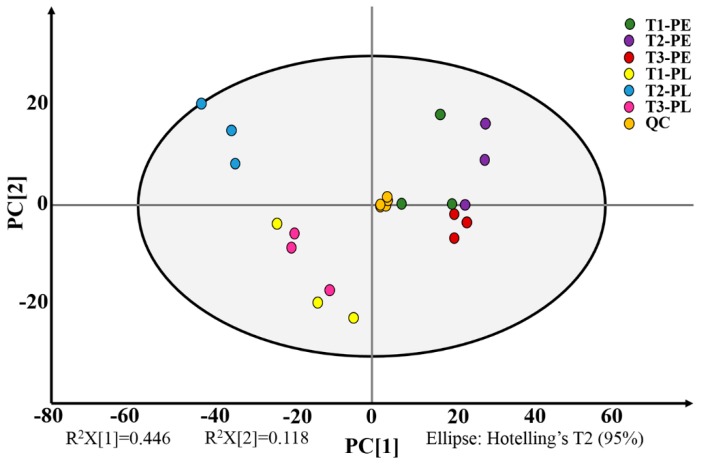
Score scatter plot for principal component analysis with quality control (QC). T1, NH_4_^+^:NO_3_^−^ = 0:100; T2, NH_4_^+^:NO_3_^−^ = 25:75; T3, NH_4_^+^:NO_3_^−^ = 50:50, PE, pericarp; PL, placenta. R^2^X: The interpretation of X variable.

**Figure 2 foods-09-00150-f002:**
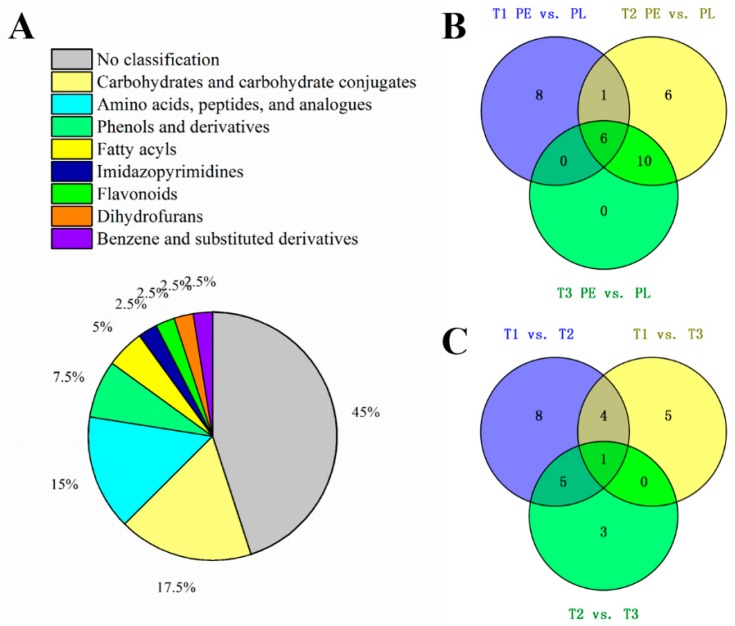
Pie chart (**A**) showing differential metabolite classification. Venn diagrams showing the shared differential metabolite numbers of different tissues (**B**) and treatments (**C**). T1, NH_4_^+^:NO_3_^−^ = 0:100; T2, NH_4_^+^:NO_3_^−^ = 25:75; T3, NH_4_^+^:NO_3_^−^ = 50:50, PE, pericarp; PL, placenta.

**Figure 3 foods-09-00150-f003:**
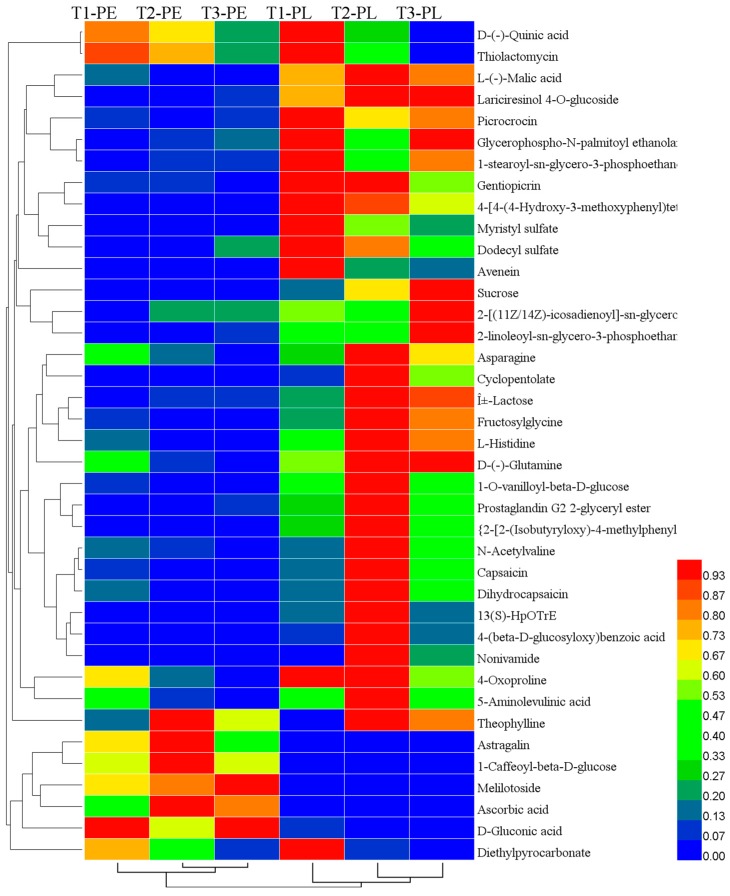
Hierarchical clustering of differential metabolites in pepper pericarp and placenta at different NH_4_^+^:NO_3_^−^ ratios. T1, NH_4_^+^:NO_3_^−^ = 0:100; T2, NH_4_^+^:NO_3_^−^ = 25:75; T3, NH_4_^+^:NO_3_^−^ = 50:50, PE, pericarp; PL, placenta. Euclidean distance and average linkage were used to construct the clustering of metabolites. The color block represents the relative expression of metabolites in the corresponding position.

**Figure 4 foods-09-00150-f004:**
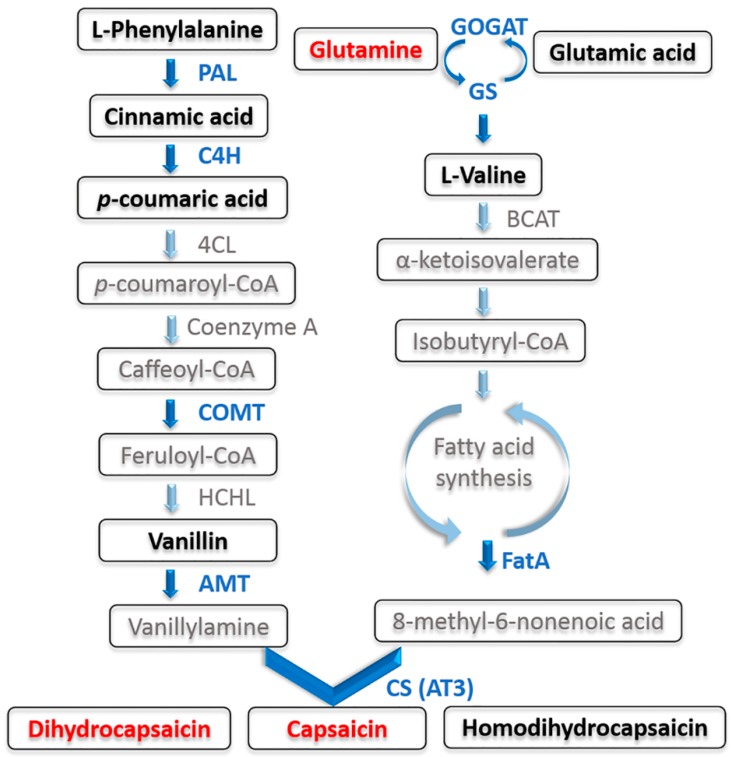
Diagram of the synthesis of capsaicinoids adapted from Mazourek et al. and Kim et al. Phenylalanine ammonia lyase (PAL), cinnamate 4-hydroxylase (C4H), 4-coumarate-CoA ligase (4CL), COMT, caffeoyl-CoA 3-O-methyltransferase; HCHL, hydroxycinnamoyl-CoA hydratase lyase; AMT, aminotransferase; GS, glutamine synthetase; GOGAT, glutamate synthase; BCAT, branched-chain amino acid aminotransferase; FatA, acyl-ACP thioesterase; CS, capsaicinoid synthetase. Non-measured and non-detected metabolites are set in gray and metabolites quantified by liquid chromatography-mass spectrometry are set in black. Significantly changed (VIP > 1 from partial least squares discriminant analysis and *p* < 0.05 from *t*-test) metabolites are set in red. Genes marked by blue arrows were analyzed in this experiment.

**Figure 5 foods-09-00150-f005:**
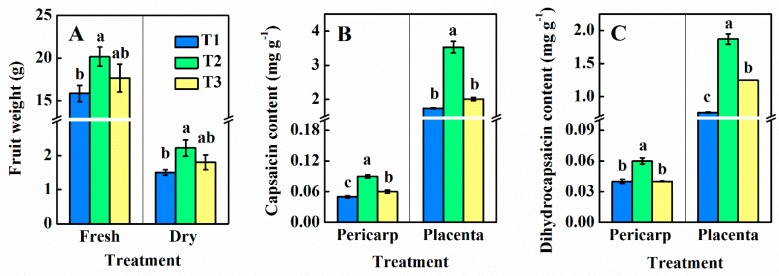
Fruit weight (**A**), capsaicin (**B**), and dihydrocapsaicin (**C**) contents in pepper pericarp and placenta at different NH_4_^+^:NO_3_^−^ ratios. T1, NH_4_^+^:NO_3_^−^ = 0:100; T2, NH_4_^+^:NO_3_^−^ = 25:75; T3, NH_4_^+^:NO_3_^−^ = 50:50. Vertical bars represent the mean ± SE (*n* = 3) and different letters denote significant differences (*p* < 0.05).

**Figure 6 foods-09-00150-f006:**
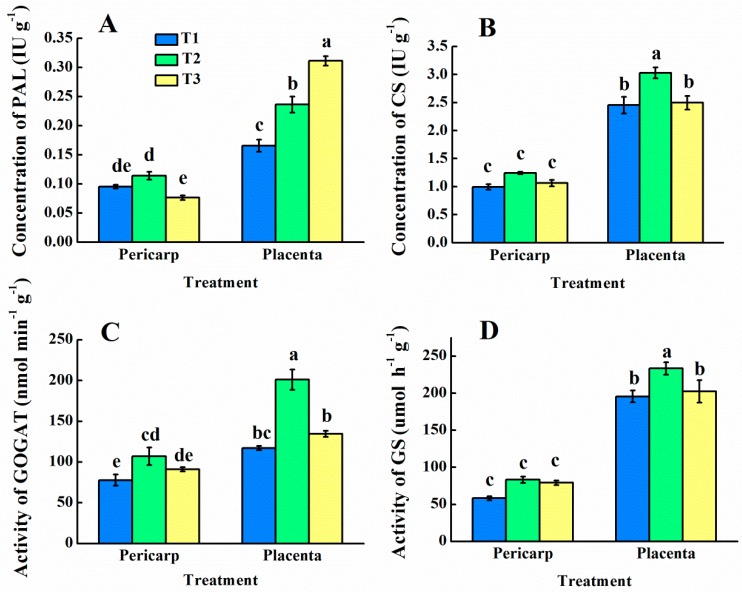
Activities of phenylalanine ammonia-lyase (PAL) (**A**), capsaicinoid synthetase (CS) (**B**), glutamate synthase (GOGAT) (**C**), and glutamine synthetase (GS), and (**D**) enzymes in pepper pericarp and placenta at different NH_4_^+^:NO_3_^−^ ratios. Vertical bars represent the mean ± SE (*n* = 3) and different letters denote significant differences (*p* < 0.05). T1, NH_4_^+^:NO_3_^−^ = 0:100; T2, NH_4_^+^:NO_3_^−^ = 25:75; T3, NH_4_^+^:NO_3_^−^ = 50:50.

**Figure 7 foods-09-00150-f007:**
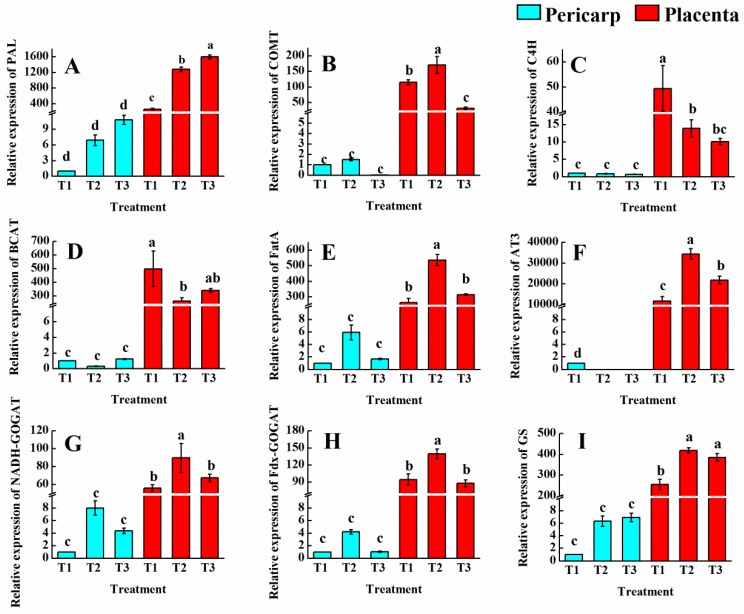
Relative expression level of genes in pericarp and placenta. (**A)**
*PAL* (phenylalanine ammonia-lyase); (**B**) *COMT* (caffeic acid-3-O-methyltransferase); (**C**) *C4H* (cinnamate 4-hydroxylase); (**D**) *BCAT* (branched-chain amino acid aminotransferase); (**E**) *FatA* (acyl-ACP thioesterase); (**F**) *AT3* (acyltransferase); (**G**) *NADH-GOGAT* (putative NADH-dependent glutamate synthase); (**H**) *Fdx-GOGAT* (putative ferredoxin-dependent glutamate synthase); (**I**) *GS* (glutamine synthetase). T1, NH_4_^+^:NO_3_^−^ = 0:100; T2, NH_4_^+^:NO_3_^−^ = 25:75; T3, NH_4_^+^:NO_3_^−^ = 50:50. Vertical bars represent the mean ± SE (*n* = 3) and different letters denote significant differences (*p* < 0.05).

**Figure 8 foods-09-00150-f008:**
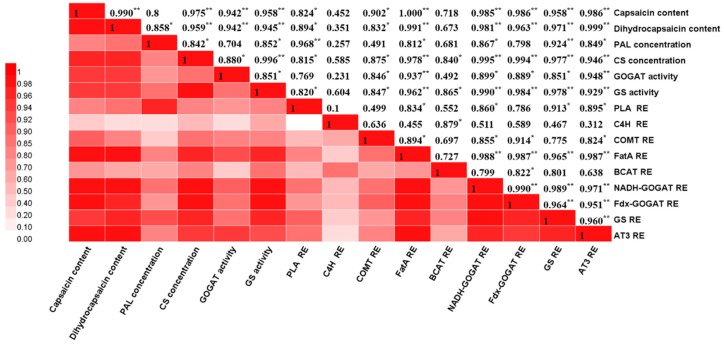
Heat map of Pearson’s correlation coefficients. PAL, phenylalanine ammonia-lyase; C4H, cinnamate 4-hydroxylase; COMT, caffeic acid-3-O-methyltransferase; BCAT, branched-chain amino acid aminotransferase; FatA, acyl-ACP thioesterase; GS, glutamine synthetase; NADH-GOGAT, putative NADH-dependent glutamate synthase; Fdx-GOGAT, putative ferredoxin-dependent glutamate synthase; AT3, acyltransferase; RE, relative expression. Values are Pearson’s correlation coefficients. * and ** denote correlation coefficients that are significant at *p* < 0.05 and 0.01 level, respectively.

**Table 1 foods-09-00150-t001:** Primers used for qRT-PCR.

Gene	Sequence (5′-3′)	Accession Number
*PAL*	F: 5′-CAACAGCAACATCACCCCATGTTTGC-3′	AF081215
	R: 5′-GCTGCAACTCGAAAAATCCACCAC-3′	
*C4H*	F: 5′-TCAGATTCCTTCCATTCGGT-3′	EU620574
	R: 5′-CTTTCTCCGTGGTGTCGAG-3′	
*COMT*	F: 5′-AAACAAGCCATAGCCTAACTCAAAC-3′	AF081214
	R: 5′-AAGTAGCAAGAAGCCTAAACATTCG-3′	
*BCAT*	F: 5′-AAAGCGTTTAGAAGAGAGGATGG-3′	AY034379
	R: 5′-GACAAGGAATGTGTACTCAGGTG-3′	
*FatA*	F: 5′-CAATGTTGTCTCGGGGGAGTTTTC-3′	AF318288
	R: 5′-CTCTCTCTCTCATTAGTAGCTACAGC-3′	
*AT3*	F: 5′-CCTCATGCATCTCTTGCAGAGAGCATAG-3′	AY819027
	R: 5′-GTCGTATGATCACGAGTAACGCTAGACC-3′	
*GS*	F: 5′-GGAAGGGACACAGAGAAGGC-3′	XM_016717075.1
	R: 5′-AACAAGCGATCCTTCGAGCA-3′	
*NADH-GOGAT*	F: ATGAATGATGACGAGGACTTTGC	EU616574
	R: GTCACGACTGTTTGCTT	
*Fdx-GOGAT*	F: TTGGGAAAGGAGTTGATGGG	EU616563
	R: AACAGCACCTACAGCAAGAAGAAT	
*Actin*	F: GTCCTTCCATCGTCCACAGG	XM_016722297.1
	R: GAAGGGCAAAGGTTCACAACA	

Note: GenBank (https://www.ncbi.nlm.nih.gov/).

**Table 2 foods-09-00150-t002:** Metabolic pathway analysis from the Kyoto Encyclopedia of Genes and Genomes (KEGG).

Treatments	Pathway	Total	Hits	Raw *p*	Impact
T1 PE vs. PL	Biosynthesis of alkaloids derived from terpenoid and polyketide	4	2	0.04	0.18
	Carbon metabolism	4	2	0.04	0.18
	Taste transduction	4	2	0.04	0.18
	Rel cell carcinoma	1	1	0.09	0.09
	Biosynthesis of secondary metabolites - unclassified	1	1	0.09	0.09
	Pathways in cancer	1	1	0.09	0.09
	Methane metabolism	1	1	0.09	0.09
	Pentose phosphate pathway	1	1	0.09	0.09
	Carbon fixation in photosynthetic organisms	1	1	0.09	0.09
	Starch and sucrose metabolism	1	1	0.09	0.09
T2 PE vs. PL	Phosphotransferase system (PTS)	4	3	0.04	0.32
	Galactose metabolism	2	2	0.05	0.24
	Carbohydrate digestion and absorption	2	2	0.05	0.24
T3 PE vs. PL	Central carbon metabolism in cancer	6	3	0.07	0.36
PE T1 vs. T2	Microbial metabolism in diverse environments	12	4	0.01	0.22
	Caffeine metabolism	1	1	0.05	0.05
	Pentose phosphate pathway	1	1	0.05	0.05
	Glycine, serine and threonine metabolism	1	1	0.05	0.05
	Biosynthesis of secondary metabolites	26	4	0.10	0.47
PL T1 vs. T2	Carbohydrate digestion and absorption	2	2	0.01	0.11
	Galactose metabolism	2	2	0.01	0.11
	Phosphotransferase system (PTS)	4	2	0.05	0.22
PE T1 vs. T3	Caffeine metabolism	1	1	0.05	0.05
PL T1 vs. T3	Carbohydrate digestion and absorption	2	2	0.04	0.04
	Galactose metabolism	2	2	0.04	0.04
	Phosphotransferase system (PTS)	4	2	0.07	0.07
PE T2 vs. T3	Pentose phosphate pathway	1	1	0.04	0.04
PL T2 vs. T3	Mineral absorption	2	1	0.04	0.04
	Alanine, aspartate and glutamate metabolism	3	1	0.05	0.05
	Cyanoamino acid metabolism	3	1	0.05	0.05
	Aminoacyl-tRNA biosynthesis	4	1	0.07	0.07
	Protein digestion and absorption	4	1	0.07	0.07

Note: KEGG (https://www.kegg.jp/kegg/pathway.html); Total, the number of metabolites in the pathway; Hit, the number of differential metabolites hitting the pathway; Raw p, *P* value obtained by enrichment analysis; Impact, impact factors of the topological analysis. T1, NH_4_^+^:NO_3_^−^ = 0:100; T2, NH_4_^+^:NO_3_^−^ = 25:75; T3, NH_4_^+^:NO_3_^−^ = 50:50, PE, pericarp; PL, placenta.

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
