# Peer review of "Nitrogen Source Affects the Composition of Metabolites in Pepper (Capsicum annuum L.) and Regulates the Synthesis of Capsaicinoids through the GOGAT–GS Pathway"

_foods, 2020, doi:10.3390/foods9020150_

Round 1
Reviewer 1 Report
This article describes the variations in the metabolite content of pepper fruits following changes in the ammonia and nitrogen input.
This is a well written article with complete methodology and observations presented. However, a mild English language editing is required to avoid misunderstanding. For eg. in line 20, ...quality formation (this sentence looks incomplete). In line 425, it should be as "With reference to pepper....
There are lengthy sentences throughout the manuscript, which needs to be concise. Also, to reduce diluting the findings from this study, the authors can reduce unnecessary referring to health and other unrelated aspects because that is not the objective of this article. At present the article more like like a review than an research report.
A strong basis for selecting the three ratios of NH4+: NO3− ratios (0:100, 25:75, and 50:50). Why 75:25 and 100:0, is not tried.
The substrates for fruit development and secondary metabolite production will be basically from the other vegetative parts of the plant. How does the change in the ammonia and nitrate ratio affects their quality and composition is missing in this article. If that data is included, it will an interesting study.
However, this is only a suggestion. If the author has a justification for not including this he can say why in the discussion and stress the novelty of this study.
Reviewer 2 Report
The manuscript is well structured, the text is easy to read, presenting current references. However, please consider the following comments:
Line 205 “2.6 Statistical analyses”, the authors report that various statistical analyses have been applied but do not refer to which data have been applied but should do so. They should also explain why a one-way analysis of variance done and which post hoc test used. The paragraph starting at line 468 (In comparison with the treatment of sole nitrate-N, t… and another capsaicinoid-Homodihydrocapsaicin has been identified (Figure 8)), as well as Figure 8, are results and should be included in "4. Results". In the results, figures 1, 2, 3, 5, 6, 7, the captions in the figures are too small which makes interpretation of the results very difficult. In Figure 6 the letters that identify significant differences are very small. In Figure 7 it is even impossible, in the printed version of the manuscript, to perceive the correlation coefficients shown. In Figure 6, the identification color of the pericarp and placenta in the graphs are missing. Line 333 “For PAL (Figure 6-A) in both the pericarp and placenta, the relative expression levels in T2 and T3 were higher than that in T1”, I think the authors should say ... "levels in T2 and T3 were significantly higher than in T1." because the same letters were assigned to the three treatments that appear with blue color, but by the SE presented, the letter assigned to T1 should be different. The letters in figures 6D, 6E, 6G, 6H and 6I with the color blue should also be checked. In Figure 8 caption, the “et al” must be It written in italics and with a period after "al", and the year (2009 and 2014) should be removed. Line 393, In the phrase: “According to previous studies, capsicum has the highest vitamin C content in vegetables.”, authors should cite the most important studies. The same on line 494 at the end of the sentence: “Previous work indicates that amino acids are primary metabolites, and therefore are not included in the capsaicin synthesis pathway.”. Line 396 remove the year after the authors' names. Line 528 the word “however” is repeated. Lines 646, 648, 673, and 726, the year of articles publication is not bold.
Reviewer 3 Report
This paper presents an extendent study on the influence of type N-fertilization (ammonium salts versus nitrates) on quality, metabolome, and expression of chosen enzymes in placenta and pericarp of one species of pepper. This is an interesting study and the results are appropriately discussed.The small comments consider:
1./ There is no need to write the common names of the chemical compounds from capital letter. It is acceptable in Figure 3, but not in the text;
2./ Fragment between lines 64 and 78 is written with different spacing;
3./PAL is involved in metabolism of phenylpropanoids not phenylpropane metabolism;
4./In lines 150-152 volume of methanol-water used for extraction should be given.
Round 2
Reviewer 1 Report
The revisions look to be conveniencing. The article can be accepted.
Reviewer 2 Report
The authors made all the suggested changes, the
the manuscript has been significantly improved and now I think it is ready for publication in Foods.